# Effect of Alkylresorcinols Isolated from Wheat Bran on the Oxidative Stability of Minced-Meat Models as Related to Storage

**DOI:** 10.3390/antiox13080930

**Published:** 2024-07-30

**Authors:** Carolina Cantele, Giulia Potenziani, Ambra Bonciolini, Marta Bertolino, Vladimiro Cardenia

**Affiliations:** 1Department of Agricultural, Forest and Food Sciences (DISAFA), University of Turin, 10095 Grugliasco, TO, Italy; carolina.cantele@unito.it (C.C.); giulia.potenziani@unito.it (G.P.); ambra.bonciolini@unito.it (A.B.); marta.bertolino@unito.it (M.B.); 2AgriForFood Chromatography and Mass Spectrometry Open Access Laboratory, University of Turin, 10095 Grugliasco, TO, Italy

**Keywords:** phenolipids, meat, wheat bran, antioxidant activity, lipid oxidation

## Abstract

Due to their antioxidant activity, alkylresorcinols (ARs) extracted from by-products could represent promising natural and innovative antioxidants for the food industry. This study tested the ability of ARs isolated from wheat bran to increase the shelf-life of minced-meat models stored at 4 °C for 9 days. Fifteen alk(en)ylresorcinols (C17–C25) were recognized by GC/MS, showing good radical-scavenging (200.70 ± 1.33 μmolTE/g extract) and metal-chelating (1.38 ± 0.30 mgEDTAE/g extract) activities. Two ARs concentrations (0.01% and 0.02%) were compared to sodium ascorbate (0.01% and 0.10%) on color (CIELAB values *L**, *a**, *b**, chroma, and hue) and oxidative stability (lipid hydroperoxides, thiobarbituric acid reactive substances (TBARS), and volatile organic compounds (VOCs)) of minced-beef samples. ARs-treated samples were oxidatively more stable than those formulated with sodium ascorbate and the negative control, with significantly lower contents of hydroperoxides and VOCs (hexanal, 1-hexanol, and 1-octen-3-ol) throughout the experiment (*p* < 0.001). However, no effect on color stability was observed (*p* > 0.05). Since 0.01% of ARs was equally or more effective than 0.10% sodium ascorbate, those results carry important implications for the food industry, which could reduce antioxidant amounts by ten times and replace synthetic antioxidants with natural ones.

## 1. Introduction

In today’s dynamism of the food industry, the search for natural antioxidants has become imperative, driven by consumers’ demands for clean labels. The search for natural antioxidants is crucial because of the need to counteract lipid oxidation, a series of ubiquitous and potentially harmful chemical reactions that occur in foods, as it poses a threat to both product quality and safety, significantly reducing the shelf life [1]. Moreover, the consumption of oxidized lipids has been associated with adverse health effects caused by the onset of oxidative stress, which leads to the development of chronic diseases [2]. In meat and meat products, the lack of antioxidants and the simultaneous presence of high amounts of PUFAs in membrane phospholipids, endogenous pro-oxidants, salt (NaCl), and free molecular oxygen greatly affect the oxidation of lipids [3]. The presence and concentration of endogenous pro-oxidants like hemoglobin and myoglobin and their accessibility to muscle-membrane phospholipids play a key role in the lipid oxidation of meat [4]. In fact, membrane phospholipids are considered the most relevant substrate for the onset of lipid oxidation reactions in muscle foods, being particularly vulnerable due to their large surface area and degree of unsaturation [4]. Additionally, in fresh ground-meat preparations, the disruption of cell membranes during processing (e.g., grinding), exposes unsaturated lipids to high concentrations of pro-oxidants, promoting lipid oxidation [3,5]. Consumers can perceive oxidation in meat through the changes that occur in its color, odor, flavor, and texture, thus affecting its acceptability [6]. For instance, the free radicals generated by the oxidized lipids can, in turn, decompose the heme pigments, causing discoloration of the meat [7]. Odor and flavor changes in meat and meat products are caused by the development of off-odors and off-flavors due to the degradation of lipids into hydrocarbons, aldehydes, ketones, acids, and alcohols [6]. Furthermore, lipid oxidation indirectly causes texture changes in meat by inducing the co-oxidation of proteins, which alters the structure and spatial arrangement of muscle proteins [6].

Wheat bran represents the principal co-product of the milling of wheat and constitutes 10–14% of the grain, with an annual production equal to 150 million tons, and is generally intended to be discarded, used for animal feed, or burned [8]. However, wheat bran is rich in interesting phytochemical compounds with high nutritional value, such as proteins, lipids, β-glucans, vitamins, and polyphenols [8]. Straddling the chemical classes of lipids and polyphenols, wheat bran is also rich in alkylresorcinols (ARs), a homologous series of amphiphilic molecules composed of a single phenolic ring and an odd-numbered alk(en)yl side chain at position 5 of the ring, whose length in wheat varies mostly from 15 to 25 carbon atoms [8,9]. ARs are especially characteristic of cereals, including rye, wheat, and barley, and are thought to be biomarkers of whole-grain wheat and rye diet and intake, since they are present in the outer layers of the kernels [10]. Antimicrobial, anticancer, neuroprotective, antilipidemic, and gut microbiota regulation properties have been described for ARs [10]. However, contradictory results are reported on the antioxidant activity of ARs. In fact, some researchers reported them as weak hydrogen/electron donors due to the hydroxyl group at the meta position failing to hamper the formation of lipid hydroperoxides in bulk oil [11,12]. In contrast, in other studies ARs retarded the formation of hydroperoxides and thiobarbituric acid reactive substances (TBARS) in vegetable oils to a greater degree than butylated hydroxytoluene (BHT), or inhibited the autoxidation of unsaturated fatty acids (linoleic and linolenic acids) [13,14]. More recently, it was found that the entire spectrum of ARs extracted from rye bran was able to inhibit lipid oxidation of oil-in-water emulsions [15], and even the individual homologs were effective in delaying the formation of both primary and secondary oxidation products in oil-in-water emulsions and bulk oil [16]. Again, Elder and colleagues [17] demonstrated that ARs were more able to counteract lipid oxidation than α-tocopherol and BHT.

Interestingly, it has been claimed by numerous studies that ARs can protect cellular lipid components from oxidative processes, being particularly effective at inhibiting Fe^2+^-induced peroxidation of fatty acids and phospholipids in a liposomal bilayer membrane [11,12,13]. Antioxidant effects have been observed for ARs in phospholipid bilayers, and this has been related to their ability to incorporate into membranes thanks to their hydrophobicity. Here, they would competitively inhibit the abstraction of hydrogens in the bis-allylic position of phospholipids, thus being more a matter of physicochemical properties and positioning rather than conventional antioxidant properties [11].

Therefore, based on these considerations, ARs could be considered as effective natural antioxidants for the protection of foods, especially muscle foods, thus meeting the need for a clean label. To the best of our knowledge, no research has been conducted on the antioxidant capabilities of ARs to counteract lipid oxidation in meat. Therefore, the present work aimed at evaluating the ability of ARs isolated from wheat bran to extend the shelf life of minced-meat model systems by delaying lipid oxidation. Specifically, two concentrations were tested, and the results were compared with those obtained in minced-meat models with sodium ascorbate added, which is the main synthetic antioxidant used by the food industry in meat and meat products for the preservation and quality enhancement.

## 2. Materials and Methods

### 2.1. Materials and Chemicals

Wheat bran (*Triticum aestivum* L.) was kindly provided by Molini Bongiovanni S.p.A. (Cambiano, Italy) and immediately freeze-dried (Lio 5P, 5Pascal, Milan, Italy), ground, and sieved at <500 μm. Wheat bran was then stored under vacuum at 4 °C until the subsequent analyses. Ground beef meat (*Longissimus dorsi* muscle) was purchased in a local market (Turin, Italy). Medium-chain triglycerides (MCT) were supplied by Ingo Steyer KG (Hemmoor, Germany).

Ethyl acetate, *n*-hexane, propan-2-ol, 1-butanol, diethyl ether, chloroform, ethanol absolute anhydrous, iso-octane, and hydrochloridric acid (37%) were purchased from Carlo Erba (Milan, Italy), while methyl tert-butyl ether (MTBE) was from VWR (Milan, Italy). The 5α-Cholestan-3β-ol, anhydrous pyridine, N,O-bis(trimethylsilyl)trifluoroacetamide with trimethylchlorosilane (BSTFA:TMCS, 99:1, *v*/*v*), ferrozine, ethylenediaminetetraacetic acid tetrasodium salt dihydrate (EDTA), 2-2′-diphenyl-1-picrylhydrazyl (DPPH), Tween^®^ 20, ammonium thiocyanate, barium chloride dihydrate, ferrous sulfate heptahydrate, sodium phosphate monobasic dihydrate, sodium phosphate dibasic anhydrous, thiobarbituric acid, 1,1,3,3-tetramethoxypropane, potassium chloride, 2,6-di-tert-butyl-4-methylphenol (butylated hydroxytoluene, BHT) and trichloroacetic acid were bought from Merk (Darmstadt, Germany). To prepare double-distilled water, a Milli-Q filter system (Millipore, Milan, Italy) was used. N° 1 and n° 41 filter papers were purchased from Whatman (Maidstone, England). Solid-phase extraction (SPE) cartridges (Strata NH_2_, 55 μm, 70 Å, 1 g/6 mL; Strata SI-1 Silica, 55 μm, 70 Å, 500 mg/3 mL) were purchased from Phenomenex (Torrence, CA, USA).

### 2.2. Alkylresorcinols Extraction and Purification

Alkylresorcinols (ARs) were extracted from wheat bran by solvent extraction and purified by solid-phase extraction (SPE) according to a procedure adapted from Esche et al. [18]. Briefly, 40 mL of *n*-hexane:chloroform (1:1, *v*/*v*) were added to 10 g of bran and left under stirring at room temperature for 1 h in darkness. After filtration, the solvent was removed with a rotary evaporator (Rotavapor, R-210, Buchi, Switzerland) at 37 °C, and the residue (lipid fraction) was dissolved in 10 mL of *n*-hexane. Afterward, a Strata NH_2_-SPE cartridge was conditioned with *n*-hexane (10 mL) and then loaded with 1 mL of the oil solution. Different solvents were then eluted to remove major interferents: *n*-hexane:diethyl ether (98:2, *v*/*v*; 10 mL), *n*-hexane:ethyl acetate (96:4, *v*/*v*; 20 mL), and *n*-hexane:ethyl acetate (5:95, *v*/*v*; 10 mL). ARs were finally eluted with 10 mL of *n*-hexane:ethyl acetate (5:95, *v*/*v*) and 5 mL of MTBE, which were combined, dried under vacuum, and dissolved in 500 μL of *n*-hexane:propan-2-ol (3:2, *v*/*v*; Fraction 1). A second SPE (Strata SI-1 Silica) was then activated with 3 mL of *n*-hexane and Fraction 1 loaded. Five milliliters of *n*-hexane:diethyl ether (8:2, *v*/*v*) and 4 mL of *n*-hexane:diethyl ether (1:1, *v*/*v*) were eluted and discarded. Three milliliters of methanol were then eluted to collect the ARs. The solvent was then evaporated, and the ARs were reconstituted with 1.5 mL of *n*-hexane:propan-2-ol (3:2, *v*/*v*; Fraction 2).

### 2.3. Characterization and Quantification of ARs

The qualitative composition of Fraction 2 was determined by gas chromatography coupled with mass spectrometry (GC/MS). Before injection, the ARs were silylated using 100 μL of pyridine and 200 μL of BSTFA:TMCS (99:1, *v*/*v*) and slowly stirred at 40 °C for 20 min. The solvents were evaporated under nitrogen, and the molecules were resuspended in 100 μL of *n*-hexane. For the identification, a Shimadzu QP2010 Plus GC/MS (Shimadzu, Kyoto, Japan) equipped with a RXi-5ms fused silica capillary column (10 m × 0.1 mm, 0.1 μm film thickness; Restek, Bellefonte, PA, USA) and an AOC-5000 Pal autosampler (Shimadzu, Kyoto, Japan) was used. One microliter of the sample was injected at 345 °C with a split ratio of 1:50, using helium as a carrier gas and with a constant linear velocity equal to 49.9 cm/s. The oven temperature was programmed from 100 °C to 310 °C (7 °C/min) and then to 320 °C (1 °C/min). The temperature was then raised to 345 °C at 7 °C/min and held for 15 min. The ion source and interface temperatures were 200 °C and 345 °C, respectively. The ions were acquired in scan mode (1166 amu/s) with a mass range of 33–700 *m*/*z*, and the mass spectra were compared to those present in the NIST08s (National Institute of Standards and Technology, Gaithersburg, MD, USA) library to identify the compounds. The extraction yield was determined by quantifying the ARs in Fraction 2. The latter, before silylation, was added to 80 μL of 5α-cholestan-3β-ol (1.005 mg/mL; internal standard) and then silylated and injected into a GC/FID (GC-2010, Shimadzu, Kyoto, Japan) equipped with the same capillary column under the same above-mentioned analytical conditions. 

### 2.4. Assessment of the Antioxidant Properties of ARs

The metal chelating ability (ferrous ion-chelating spectrophotometric assay) and radical-scavenging activity (DPPH• spectrophotometric assay) were assessed to define the antioxidant properties of the ARs extracted and purified from wheat bran, according to the procedures described elsewhere [19]. For the tests, the extracted ARs were resuspended in ethanol, and the absorbance was measured using a BioTek Synergy HT spectrophotometric multi-detection 96-well microplate reader (BioTek Instruments, Milan, Italy). To quantify the radical scavenging and ferrous ion-chelating activities, calibration curves were constructed using Trolox (25–30 μM; y = 0.1941x + 0.0483; R^2^ = 0.9999) and EDTA (0.001–0.1 mg/mL; y = 8546.5x − 0.4859; R^2^ = 0.9947), respectively. Micromoles of Trolox equivalents per gram of ARs extract (μmolTE/g) and milligrams of EDTA equivalents per gram of ARs extract (mgEDTAE/g) were used to express the results.

### 2.5. Preparation of Minced-Meat Models and Storage Conditions

To deliver the ARs into the minced-meat models, 1.0% oil-in-water (O/W) emulsions were prepared and subsequently mixed with the ground meat. To prepare the emulsions, 1.0% (*w*/*w*) medium-chain triglycerides (MCT), 0.1% (*w*/*w* oil) Tween 20, and 10 mM phosphate buffer solution (pH 7) were used. ARs were dissolved in the oil by stirring at room temperature after the removal of the solvent. The emulsion was roughly homogenized using an IKA T25 digital Ultra-Turrax^®^ (IKA^®^-Werke GmbH & Co. KG, Staufen, Germany) for 2 min at 20,000 rpm and then refined with a VCX750 ultrasonic processor (Sonics & Materials, Inc., Newton, CT, USA) equipped with a ½ inch diameter tip probe operating at an amplitude of 50%. During sonication, the emulsions were kept in an ice bath to prevent the temperature exceeding 30 °C. To evaluate the effect of the ARs on their shelf-life, five different minced-meat samples were prepared: (1) C_NEG, negative control, with emulsion without antioxidants; (2) AR01, with emulsion added with ARs at 0.01% (*w*/*w* of total meat); (3) AR02, with emulsion added with ARs at 0.02% (*w*/*w* of total meat); (4) C_POS+, positive control, with emulsion added with sodium ascorbate at 0.01% (*w*/*w* of total meat); and (5) C_POS++, positive control, emulsion added with sodium ascorbate at 0.10% (*w*/*w* of total meat). Sodium ascorbate was used at 0.01% as a positive control for direct comparison with natural antioxidants and at 0.1% to reflect the typical industry concentration in meat products. Each type of emulsion was added to ground beef at a ratio of 200 mL/kg meat and thoroughly mixed to obtain a homogeneous mixture. Samples were made with 30 ± 1.0 g of the latter and placed in sealed food contact containers in polyethylene terephthalate (PET) on a poly-coated (PE paper) paper foil at 4 ± 0.1 °C in darkness for 9 days. Three independent batches were made for each type of sample (n = 3). The first measurement (day 0) was taken after 4.5 h following the preparation of the samples.

### 2.6. Color Measurement

The color of the samples was monitored during the experiment to assess if the ARs affect this parameter and its stability during the storage. A CM-2600 spectrocolorimeter (Konica Minolta Sensing Inc., Osaka, Japan) equipped with an aperture mask of 8 mm, D65 illuminant, and 10° of Standard Observer was used to determine the CIELAB color space indices (*L**, *a**, *b**). The hue angle (*h_ab_*, expressed as degrees) and chroma (*C*_ab_*) were calculated according to Cantele et al. [20]. Six measurements were carried out for each patty, and the specular component excluded (SCE) values were considered.

### 2.7. Lipid Extraction

A modified version of the method described by Folch and colleagues [21] has been used to extract the lipid fraction from the minced beef meat. Twenty grams of each sample were added to 200 mL of *n*-hexane–propan-2-ol solution (3:2, *v*/*v*) with 0.02% BHT and homogenized for 3 min at 20,000 rpm by a T-25 Ultra-Turrax^®^ (IKA^®^-Werke GmbH & Co. KG, Staufen, Germany) in a screw-capped glass bottle. The bottle was then kept at 60 °C for 15 min, with 100 mL of *n*-hexane added, and mixed again by Ultra-Turrax^®^ for 2 min at the same speed. The solid residue was eliminated by filtration with a Buchner funnel and filter paper (Whatman n° 1). The filtrate was added with 100 mL of a 1 M KCl solution, mixed thoroughly, and left overnight at 4 °C to obtain a phase separation. Through a separating funnel, the upper phase (organic phase) was collected, added with sodium sulfate, left in the fridge for 2 h, filtrated, and dried with a rotary evaporator (Rotavapor, R-210, Buchi, Flawil, Switzerland) at 40 °C.

### 2.8. Determination of Lipid Hydroperoxides

The content of lipid hydroperoxides in the samples was determined according to Funaro et al. [22]. Twenty milligrams of the extracted lipids were added to 9.8 mL of a chloroform/methanol solution (2:1, *v*/*v*) and 50 μL of both ammonium thiocyanate and FeCl_2_ solutions, vortexed for 30 s, and incubated in darkness for 5 min. The absorbance was read at 500 nm with a UV-1800 spectrophotometer (Shimadzu, Kyoto, Japan) and quantification was achieved by building a calibration curve in the range of 1–40 μg/mL of Fe^3+^. The results were expressed as milliequivalents of oxygen per kilogram of fat (meqO_2_/kg fat).

### 2.9. Determination of Thiobarbituric Acid Reactive Substances (TBARS)

Thiobarbituric acid reactive substances (TBARS) were measured according to Tarladgis et al. [23]. Briefly, 2 g of each sample were added to 8 mL of phosphate buffer aqueous solution (pH 7) in a 50 mL Sovirel bottle with a screw cap and homogenized by Ultra-Turrax^®^ for 30 s at 21,500 rpm. Two milliliters of a trichloroacetic solution (30%, *w*/*v*) were added, and the mixture was homogenized for 30 s at 17,500 rpm. After filtration (Whatman paper filter No. 41), 5 mL of the filtrate were collected in a 20 mL Sovirel tube, and 5 mL of a TBA aqueous solution (0.02 M) were added. The capped tubes were put at 90 °C in a water bath for 20 min, then at 4 °C for 30 min, and finally sonicated for 5 min. The absorbance of the complex was read at 530 nm, and by a calibration curve built with 1,1,3,3-tetramethoxypropane in the range of 0.03–2.26 μg/mL, the quantification of malondialdehyde (MDA) was carried out. The results were expressed as milligrams of malondialdehyde (MDA) per kilogram of meat (mg MDA/kg meat).

### 2.10. Determination of Volatile Organic Compounds (VOCs)

Volatile organic compounds (VOCs) were determined through headspace solid-phase microextraction (HS-SPME) and GC/MS (QP-2010 Plus, Shimadzu, Kyoto, Japan) according to Botta et al. [24], with slight modifications as follows. The samples (2.0 g) were accurately weighed in 20 mL headspace vials equipped with aluminum caps sealed with a PTFE silicone septum. To isolate the VOCs, a DVB/CAR/PDMS-coated fused silica fiber (10 mm length, df 50/30 μm; Supelco, Bellafonte, PA, USA) was exposed to the headspace for 30 min at 40 °C after having equilibrated the vials at the same temperature for 15 min. A Combi Pal system (CTC Analytics AG, Zwingen, Switzerland) was used. The fiber was subsequently desorbed into the GC/MS inlet at 260 °C for 5 min, with a split ratio of 1:25. Chromatographic separation of the volatiles was achieved with an RTX-5 fused silica capillary column (20 m × 0.10 mm × 0.10 μm; Restek, Bellafonte, PA, USA) held for 4 min at 40 °C, then risen to 220 °C (4 °C/min) and 260 °C (20 °C/min). The final temperature was maintained for 3 min. Helium was used as a carrier gas with a constant linear velocity of 34.7 cm/s. The ion source temperature was 200 °C, while the interface temperature was 230 °C. Ions were acquired in scan mode in the range of 33–350 *m*/*z* with a scan speed of 1111 amu/sec (0.30 scan/s). Compounds were identified by comparing their spectra with those reported in the NIST08s (National Institute of Standards and Technology, Gaithersburg, MD, USA) library. Environmental contamination was avoided by blank injections of the fibers and vials. VOCs were quantified using calibration curves of the corresponding compound (hexanal, y = 222,252x + 102,461, R^2^ = 0.9981; 1-hexanol, 215,982x + 110,441, R^2^ = 0.9975; 1-octen-3-ol, y = 124,289x + 8675, R^2^ = 0.9976), and the results were expressed as mg/kg meat.

### 2.11. Statistical Analysis

All the results are presented as the mean ± standard deviation of three independent replicates (n = 3). A one-way analysis of variance (ANOVA) and Tukey’s post hoc test were used to ascertain any statistical difference between the samples with a confidence level equal to 95%. The Shapiro–Wilk and Levene’s tests were used to inspect, respectively, normality and homoscedasticity before executing the ANOVA. Data variability was explored by computing a principal component analysis (PCA) on all datasets. All the statistical analyses were performed with IBM SPSS statistical software (version 28; IBM, Chicago, IL, USA).

## 3. Results

### 3.1. Characterization of the ARs in the Purified Extract

The purified extract of wheat bran was analyzed using GC/MS, revealing the presence of fifteen distinct resorcinolic lipids, including seven saturated alkylresorcinols (C15:0, C17:0, C19:0, C21:0, C23:0, C25:0, and C27:0) and ten monounsaturated alkenylresorcinols (C17:1, C19:1, C21:1, C23:1, and C25:1, with two isomers each) (Appendix A). Compared to the saturated ARs, monounsaturated ARs were found at lower concentrations (<10% of the total ARs). TMS derivatives of C15:0, C17:0, C19:0, C21:0, C23:0, C25:0, and C27:0 were identified using molecular ions at *m*/*z* 464, 492, 520, 548, 576, 604, and 632, respectively (Appendix A). On the other hand, the molecular ions of the corresponding alkenylresorcinols were identified at *m*/*z* 490, 518, 546, 574, and 602, respectively. Additionally, the characteristic peaks at 268 *m*/*z* and 281 *m*/*z*, produced by the McLafferty rearrangement during the fragmentation of ARs, were used as qualifier ions for all the alk(en)ylresorcinols (Appendix A). The percentage composition of ARs was 0.48 ± 0.03% (C15); 7.60 ± 0.04% (C17); 42.38 ± 0.15% (C19); 39.43 ± 0.26% (C21); 7.57 ± 0.13% (C23); 2.10 ± 0.20% (C25); and 0.44 ± 0.02% (C27). An extract with 85.70 ± 0.30% purity was obtained; the remaining contaminants were stearic acid, palmitic acid, glycerol monostearate, and 1-monopalmitine. The extraction yield of the alkylresorcinols was 471.77 ± 32.97 μg/100 mg of lipid fraction extracted from wheat bran, which corresponded to 13.27 ± 0.66 mg/g of wheat bran.

As for the antioxidant properties, the extract was found to have an antiradical capacity equal to 200.70 ± 1.33 μmol TE/g extract and an ability to chelate ferrous ions equal to 1.38 ± 0.30 mg EDTAE/g extract.

### 3.2. Influence of ARs on the Visual Appearance

The effect of storage, as well as of the addition of sodium ascorbate and ARs and their concentrations, on the CIELAB parameters *L**, *a**, *b**, hue (*h_ab_*), and chroma (*C*_ab_*) in the minced-meat models is shown in Table 1. The color was measured in all samples immediately after their preparation, and no significant (*p* > 0.05) differences were observed between them for *L**, *a**, and *h_ab_*, while *b** and *C*_ab_* were affected by the presence of sodium ascorbate. In fact, on day 0, the color coordinate *b** was higher in both C_POS+ and C_POS++ (*p* < 0.05). As the chroma defines the color intensity (saturation), the presence of sodium ascorbate in the meat during sample preparation led to a less greyish color. Afterward, only C_POS++ was able to keep these parameters more stable over time. In contrast, C_POS+ and both concentrations of ARs had no effect on color compared to C_NEG (*p* > 0.05). During the storage, color coordinate *L** significantly increased in all samples except for C_POS++, where it remained constant, except on day 6 when it dropped and then rose again on the last day to return to the initial value. Indeed, on each sampling day, the *L** in C_POS++ was consistently lower than in all other samples (*p* < 0.05), which, in contrast, showed no significant difference from each other (*p* > 0.05). Thus, all samples except C_POS++ became lighter during storage. Regarding *a**, in all samples, this value decreased significantly during the experiment (*p* < 0.01). However, again, C_POS++ showed the mildest (from 10.58 ± 0.58 to 8.61 ± 0.96) and most gradual decrease (at day 3 it was still no different from day 0; *p* > 0.05), implying that the samples were redder. The colorimetric value *b** also changed throughout the cold storage in all samples except for C_POS++. Specifically, yellowness increased from 14.18–15.11 to 16.10–16.68. The saturation of all the samples except for C_POS++ again decreased significantly until day 6 and increased on the last day of the experiment. Finally, during the cold storage, the *h_ab_* increased significantly (*p* < 0.001) in all samples, including C_POS++, going from 54.18–55.58 to 61.67–67.07. However, C_POS++ reached the lowest *h_ab_* value at the end of the storage (*p* < 0.001).

### 3.3. Influence of ARs on the Oxidative Stability

#### 3.3.1. Primary Oxidation Products

The assessment of lipid hydroperoxides, whose results are presented in Figure 1 (and detailed in Table A1 of Appendix B) revealed distinctive trends among the tested samples, but in all cases, a significant increase after the first measurement was observed (*p* < 0.001).

During the storage, C_NEG and C_POS+ displayed higher hydroperoxide content than the other samples (*p* < 0.001), indicating a significant influence on oxidation. After 4.5 h (day 0), the level of hydroperoxides was already higher in C_POS+ (1.26 ± 0.08 meqO_2_/kg meat) than in the other samples, while AR02 was characterized by the lowest amount (0.67 ± 0.10 meqO_2_/kg meat). Higher values in C_NEG and C_POS+ at the first measurement indicate a more marked triggering of oxidation reactions by lipid contact with oxygen than in the other treatments, where the presence of antioxidants evidently hampered them. As storage progressed, C_POS+ revealed the same trend shown by C_NEG, with overlapping peroxide values (*p* > 0.05). In fact, on day 3 of storage, the maximum level of hydroperoxides observed in the experiment was reached (5.69 ± 0.03 meqO_2_/kg meat and 5.35 ± 0.05 meqO_2_/kg meat for C_NEG and C_POS+, respectively). A decrease in their values was then observed on day 6 and day 9, corresponding to the degradation of these compounds to form the secondary oxidation compounds, including aldehydes and ketones. Completely different behavior was displayed by the samples added with ARs and 0.10% sodium ascorbate. In fact, significantly lower values were recorded throughout the experiment (*p* < 0.001), reaching their highest contents only on the last day. Most interestingly, the content of hydroperoxides in AR01 and C_POS++ was the same throughout the experiment (*p* > 0.05), highlighting how ARs at 0.01% retarded the formation of these compounds as effectively as sodium ascorbate at the concentration normally used by industries. The incorporation of ARs at 0.02% overtakes the efficacy of C_POS++ (*p* < 0.001), demonstrating a remarkable reduction in hydroperoxide formation throughout storage. These findings highlight the potential of ARs, particularly at 0.02%, to be effective agents for inhibiting the development of primary oxidation products in minced-meat models.

#### 3.3.2. Secondary Oxidation Products

In the present study, a time-dependent increase in TBARS was observed in all samples until day 3. Meanwhile, on day 6, the values decreased, except for C_NEG and C-POS++, where the decrease was delayed for another 3 days (Figure 2 and detailed in Table A1 of Appendix B). In fact, the initial values of the 0.24–0.68 mg MDA/kg meat (range of all samples) reached at day 6 were 0.50–1.07 mg MDA/kg meat and then decreased to 0.42–0.65 mg MDA/kg meat at day 9. 

C_NEG showed the highest values, which were double or triple that of the other treatments, except for C_POS+, which again showed a similar trend to the negative control (*p* > 0.05). Both concentrations of ARs successfully restrained the formation of TBARS with respect to C_NEG and were demonstrated to be more effective than C_POS+ and C_POS++. Already after 4.5 h, the TBARS in C_NEG (0.34 ± 0.05 mg MDA/kg meat) were slightly higher than the other samples (range 0.24–0.29 mg MDA/kg meat), although it was not statistically significant (*p* > 0.05). After 3 days of storage, the content of TBARS in C_NEG and C_POS+ dramatically increased, reaching 0.95 ± 0.22 mg MDA/kg meat and 1.00 ± 0.10 mg MDA/kg meat, respectively, while in the other treatments, TBARS doubled with respect to the first measurement (4.5 h). At day 6, TBARS continued to increase in C_NEG and C_POS++, while in AR01 and AR02, they remained constant. On the last day, AR02 showed an even lower content than C_POS++ (*p* < 0.001). 

In the present study, along with TBARS, VOCs were also monitored. Twenty-six different volatile organic molecules were identified in the samples, including carbon dioxide, ethanol, carbon disulfide, pentane, acetic acid, 3-methyl-butanal, 2-methyl-butanal, 1-penten-3-ol, 3-pentanone, acetoin, 3-methyl-1-butanol, 1-pentanol, 2,3-butanediol, 2,4-dimethyl-hexane, hexanal, 2-octene, 1-hexanol, 1-octen-3-ol, 3-methyl-butanoic acid, 2-heptanone, 2,2-dimethyl-3-heptanone, 2,2,4,6,6-pentamethyl-heptane, 2,3-octanedione, 2-pentyl-furan, 2,6-dimethyl-undecane, and dodecane. Only three, among the principal markers of lipid oxidation, exhibited clear and significant differences between the treatments and are thus reported: hexanal, 1-hexanol, and 1-octen-3-ol (Figure 3). The effect of ARs on the development of these secondary products appears evident. In fact, when ARs were present in the samples at both concentrations (0.01% and 0.02%), a significantly lower value with respect to the control was recorded for the three molecules from the first sampling (*p* < 0.05). Specifically, hexanal was already found 4.5 h after the preparation of the samples in C_NEG. In C_POS+ hexanal was found from day 3, increasing up to day 6 and then decreasing, and in C_POS++ only from day 6, again decreasing on the last day. AR01, on the other hand, was characterized by the presence of hexanal only from day 6 on, and with significantly lower values than in all the other samples (*p* < 0.05). Interestingly, the AR02 sample never showed any development of this aldehyde during its shelf life. High levels of 1-hexanol were observed in both C_NEG and C_POS+ throughout the storage, with an increasing trend until day 6. In the other treatments, 1-hexanol was found in low amounts but was still significantly lower in AR02 than in AR01 and C_POS++ (*p* < 0.05). Similar results were also found for 1-octen-3-ol, whose content in AR01 and AR02 was the lowest compared with the other treatments during the experiment. C_NEG again displayed the highest values throughout the storage, followed by C_POS+. C_POS++ was shown to be not as effective as AR01 and AR02, where 1-octen-3-ol was detected at the lowest levels at the end of the experiment (*p* < 0.001).

### 3.4. Principal Component Analysis (PCA)

To highlight the relationships between the identified factors and the samples of beef burgers, multivariate analysis was employed. According to PCA, the first two principal components (PC1, 42.97%; PC2, 21.74%) accounted for 64.71% of the variance among the samples. As reported in Figure 4, a clear separation of VOCs from the other measured parameters was observed. Again, the components used to describe the color were also completely separated, where *a** was mainly correlated to PC1 and inversely correlated to *b**, which resulted in mostly correlated to PC2. On the other hand, chroma (*C*_ab_*) was inversely correlated with VOCs. In addition, *L** and hue angle (*h_ab_*) were positively correlated with PV and TBARS, since as peroxides increased, significant changes in the lightness of the color were observed. Moreover, as reported in Biplot (Figure 4), different clusters are recognized. As expected, all fresh samples (T0) were clustered together, and no effect due to the presence of ARs was observed; in addition, the samples were mainly characterized by *a** component, since the red color was equally present. TBARS and PV, as well as *L**, largely categorized negative control, ARs at low concentration, and positive control at a low dosage sample at day 3 of storage (T3). The T6 control samples, both negative and positive at low dosage, were classified by VOCs, which confirms that, after 6 days of storage, the peroxides were degraded, leading to an increase in the formation of VOCs. At T9 of storage, only the control at a high dosage of sodium ascorbate (C_POS++), together with samples containing ARs, were clustered together. These results confirm that ARs could be considered a valuable and natural alternative to ascorbic acid in the formulation of beef burgers, contributing to reducing the environmental impact linked to the use of synthetic preservatives.

## 4. Discussion

In the present work, alkylresorcinols (ARs) were isolated from wheat bran and chemically characterized, and their antioxidant properties in vitro were assessed. Then, the capability of the ARs to counteract lipid oxidation was tested in minced-meat model systems as related to storage.

The composition of the AR extract was in line with the literature [9], with C19 and C21 as the main ARs, followed by C17, C23, C25, C15, and C27. While the overall content of ARs varies significantly even within the same species, due to genetic, environmental, and agronomic factors, the distribution of the homologs within species is relatively consistent. For example, the C17:0/C21:0 ratio for durum wheat is equal to ~0.01, ~0.1 for soft and spelt wheat, and ~1.0 for rye [25]. In the present study, the C17:0/C21:0 ratio was found to be 0.19, in agreement with the literature.

Free radicals and bivalent cations like Fe^2+^ and Cu^2+^ represent the main initiators of lipid oxidation reactions, thus constituting the main concern in lipid-containing foods. Therefore, the purified extract of ARs was tested to evaluate if it was either a free radical scavenger and/or metal ion chelator capability able to retard lipid oxidation. The search for similarities in the literature for in vitro antioxidant test results is never easy or obvious, as differences in operating conditions make comparisons often misleading. Indeed, conflicting results are reported in the literature, with works reporting little or no ability of alkylresorcinols to donate a hydrogen/electron [11,12], while others claim good antiradical activity [13,14,15,17]. It is true, however, that the chemical structure of resorcinol does not in itself suggest a high activity, as the hydroxyl group occurs at the *meta*-position, which is notoriously deleterious to antioxidant activity [11]. Despite this, we found good antiradical activity comparable, if not greater, than that found in the literature for wheat-bran extracts rich in ARs or individual homologs [26]. In addition, a positive response in the chelating-activity assay was observed despite the absence of a catechol or galloyl group on the benzene ring of the resorcinolic lipids. This phenomenon has also been noted in other molecules, such as ferulic acid, where chelating activity is attributed to alternative interactions with iron [19,27]. After confirming the antioxidant properties of the ARs through the previously described in vitro assays, the purified extract was added to minced-meat models, where its effectiveness in counteracting lipid oxidation and its impact on color were then compared to those of sodium ascorbate.

Color is the first sensory attribute that consumers perceive in food and ultimately determines their propensity to purchase or reject it [28]. Specifically for meat products, it is determined by the amount of myoglobin present at the surface of the meat and its chemical state, the pH-dependent structure of the muscle, and the marbling [29]. Many factors throughout the meat production chain (farmers, processors, and consumers) affect meat color. Pre-harvest factors include type of diet, feeding types, energy intakes, withdrawal times, pre-slaughter stress, and slaughter techniques, while post-harvest factors include storage temperature, packaging conditions such as materials and atmosphere compositions, and lipid and protein oxidation [28,30]. One of the solutions that can be adopted to keep the color of the meat stable over time is the addition of sodium ascorbate. Ascorbate is a reducing agent that not only inactivates free radicals inhibiting lipid oxidation but also maintains the reduced ferrous state of myoglobin when added to fresh muscle foods [31]. Although the European Food Safety Authority (EFSA) established that there are no genotoxicity and safety concerns for this antioxidant and that it can be used as a food additive [32], sodium ascorbate is, however, synthetic, and therefore disliked by consumers. As ARs have been demonstrated to possess antioxidant properties, they could also influence the oxidative state of myoglobin and thus the color stability of the samples.

In this study, only sodium ascorbate was able to keep the color of the samples stable during cold storage and only at the highest concentration. In fact, during the experiment, all the measured parameters (*L**, *a**, *b**, *C*_ab_*, and *h_ab_*) markedly changed in all samples, while in C_POS++, the change was milder. Specifically, *L**, *b**, and *h_ab_* increased, while *a** and *C*_ab_* decreased over time. It is likely that the increase in *L** values during meat storage was caused by the enzymatic breakdown of proteins, which weakens protein structure and increases light dispersion [30]. Other authors have reported the same increase in lightness [30,33,34,35]. The decrease in *a** value is due to the oxidation of myoglobin [36,37]. In fact, when exposed to oxygen, the free binding site of the heme of myoglobin (deoxymyoglobin, purple–red color) covalently binds molecular oxygen to form oxymyoglobin, which is characterized by a bright cherry-red color. These two redox states of myoglobin have the heme iron in the ferrous state (reduced). When the heme iron of oxymyoglobin is oxidized to the ferric state, metmyoglobin is formed, which is brown in color [36,37]. Interestingly, in C_NEG, C_POS+, AR01, and AR02 samples, the *a** value increased on the last day. This phenomenon could be explained by the reduction of metmyoglobin to deoxymyoglobin, which was then rapidly converted to oxymyoglobin [36,38,39], and it can result from two circumstances. First, brown metmyoglobin can be reduced to purple reduced myoglobin by the activity of some enzymes that are present in the muscle, often called metmyoglobin-reducing activity or MRA. The process involves the donation of an electron by the reducing enzymes to the iron molecule that goes from the ferric state (Fe^3+^) to the ferrous state (Fe^2+^) [39]. Second, it can derive from bacterial growth. High enough concentrations of bacteria can prevent oxygen from reaching the surface of the meat, allowing brown metmyoglobin to be rapidly enzymatically reduced to purple myoglobin. [39]. In addition, certain types of bacteria can produce reducing equivalents that might reduce metmyoglobin back to deoxymyoglobin as well [39]. The colorimetric value *b** also changed throughout the cold storage, increasing in all samples except for C_POS++. The increase in the *b** value corresponds to an increase in yellowness, and these results are consistent with Ebrahimi et al. [34]. The saturation of all samples except for C_POS++ decreased significantly; it has been reported that a diminishing of *C*_ab_* is correlated to brown color [29]. Finally, during the cold storage, the *h_ab_* increased significantly in all samples, including C_POS++, and Lee and colleagues [31] noticed the same increment in antioxidant-treated ground-beef patties during a 6-day experiment at 4 °C, as well as Ijaz and co-workers [30] on a 7-day storage of beef.

Despite not affecting the visual appeal and being directly perceivable like the color, oxidative stability is also crucial to consider during the cold storage of beef patties, as it significantly influences consumer acceptability and food safety. Indeed, lipid oxidation is one of the main issues that deteriorate meat and meat products, decreasing their shelf-life due to their high fat content [6]. Therefore, antioxidants can be used to delay lipid oxidation and extend the product’s shelf-life [3]. ARs from wheat bran can represent an effective alternative to synthetic antioxidants. The ARs were demonstrated to be highly effective in retarding lipid oxidation in minced-meat models, restraining the formation of both primary and secondary oxidation products. Their highest concentration (0.02%) was revealed to be even more efficient than sodium ascorbate at 0.10%, and this was already evident with the formation of hydroperoxides, where AR02 displayed the lowest values for the entire experiment. This ability to inhibit the initial stages of lipid oxidation can be attributed to the antioxidant properties of ARs confirmed in this study, especially the radical-scavenging activity. Other researchers found similar results in beef patties with a phenol-rich extract obtained from olive mill wastewaters added at the same concentrations (87.5–175 mg of phenols/kg meat) [5]. As oxidation progresses, the hydroperoxides decompose, generating secondary oxidation products, one of the main ones being the aldehyde 1,3-propanedial, also called malondialdehyde (MDA), which is formed from polyunsaturated fatty acids [34]. The efficacy of ARs was also confirmed by TBARS, where the lowest values were again found in the presence of these molecules. It was also interesting to note that, in the negative control and samples with sodium ascorbate, a decrease in TBARS was only observed on the last day of the experiment. The decrease in the values of TBARS during storage has already been recorded in other studies and may be associated with the decomposition into tertiary oxidation products or from their reaction with proteins [34,40]. In addition, since the decomposition of the primary oxidation products gives rise to the secondary products, it is not surprising to see the trend of TBARS follow that observed for hydroperoxides. Indeed, for C_NEG and C_POS+, a peak in hydroperoxides was seen at day 3, which is then reflected in the increased formation of thiobarbituric acid-reactive compounds, and then declines as to both parameters. This overlap in the trend of the two oxidative parameters indicates that the formation of TBARS occurred very rapidly following the formation of hydroperoxides. In samples AR01 and AR02, due to the limited formation of hydroperoxides, there was a less pronounced increase in TBARS levels and no subsequent decrease, indicating significantly reduced oxidation. In the case of C_POS++, although an increase in hydroperoxides was not observed, an increase and then a decrease in TBARS was detected, and this phenomenon might be explained by the high instability of the primary oxidation products. 

Considering that 1 mg MDA/kg meat is reported as the threshold above which rancid flavor begins to develop in meat [5], we can conclude that ARs successfully exerted a protective effect for inhibiting the formation of rancid off-flavors in samples during cold storage, especially AR02, which never exceeded 0.57 ± 0.07 mg MDA/kg meat. Similar results were found by other researchers [41] in beef-meat hamburgers added with rye and wheat acetone and methanol extracts and stored in modified atmosphere packaging (30% CO_2_, 70% O_2_) at 4 °C in darkness. The authors concluded that adding bran extracts to meat products could be an appealing approach to improve their oxidative stability, enriching also the product with functional ingredients that have health benefits [41]. Natural extracts from other by-products have also shown greater efficacy than ascorbic acid in limiting the formation of TBARS, such as in the work of Andrés and colleagues, where aqueous extracts of red grape and olive by-products were investigated [42], or in that of Soriano and colleagues, where oak-wood extracts almost completely prevented the formation of TBARS in comparison with the positive control [35]. Lipid oxidation is a complex process that can result in a wide variety of compounds from different pathways. Among these, secondary oxidation products include different volatile organic compounds (VOCs) with different molecular structures, such as aldehydes, ketones, alcohols, carboxylic acids, and hydrocarbons [6]. In the present study, hexanal, 1-hexanol, and 1-octen-3-ol, three of the main lipid oxidation markers identified, made clear the effectiveness of ARs in comparison with other treatments [6,43,44]. The type of volatile compound generated during oxidation depends on the fatty acid composition of the meat. It is well known that hexanal constitutes one of the main aldehydes generated during lipid oxidation, and it is mainly derived from the decomposition of alkyl hydroperoxides of linoleic acid [45]. The aldehydes produced can be further decomposed into other molecules by other oxidation products or by enzymes present in the system. It has been reported that hexanal might be reduced to hexanol by intracellular enzymes, like alcohol dehydrogenases [46]. Alcohols can also be formed during lipid oxidation by the reaction of alkyl radicals with hydroxyl radicals [47]. However, it might be pointed out that 1-hexanol was found in samples where hexanal was not detected, which suggests that it might also arise from other pathways than the enzymatic and oxidative degradation of hexanal. In any case, the effectiveness of ARs, especially at the highest concentration, to restrain its development is confirmed and aligns with the results obtained for the other oxidative parameters. The 1-octen-3-ol can derive from the intermolecular cyclic rearrangement and cleavage of linoleic acid esters [48] or from the oxidative breakdown of linoleic acid catalyzed by lipoxygenases [49]. Again, samples treated with ARs showed significantly lower levels of this alcohol than the other samples, including C_POS++. For all the VOCs reported, a decrease in their levels was found on the last day in all the samples, and this could be due to advanced oxidation reactions, leading to their degradation and transformation into other compounds [47], or to the interaction of these compounds with other compounds in the matrix, or even to their metabolism by microorganisms. These results confirm the effectiveness of ARs in reducing oxidation already found for hydroperoxides and TBARS.

To the best of our knowledge, the present paper is the first to present the results of adding alkylresorcinols extracted from wheat bran to a minced-meat system. The promising results demonstrated that those compounds could be effective natural antioxidants for the food industry. This approach simultaneously addresses the demand for clean labeling and environmental sustainability, valorizing a by-product generated in large quantities every year. However, the limitations of this study and potential next steps should be highlighted. This preliminary work focused on screening the potential antioxidant activity of alkylresorcinols in minced beef. Future research should aim to develop an environmentally sustainable extraction and isolation method for these compounds, avoiding the use of toxic solvents. In addition, it is important to evaluate the effect of ARs from a microbiological point of view in order to define whether alkylresorcinols can also have an antimicrobiological effect. Lastly, after ensuring a food-grade extraction process, it will be important to conduct a study on sensory characteristics. While it has been verified that these compounds do not alter the appearance of the meat before cooking, it is also crucial to assess whether they affect the flavor and taste after cooking.

## 5. Conclusions

In the present study, the efficacy of natural alkylresorcinols (ARs), isolated from wheat bran, on the increase of minced-meat model systems’ shelf life during cold storage as alternatives to chemical preservatives was investigated. Sodium ascorbate at the lowest concentration (0.01%) was found to be completely ineffective for preserving the beef ground meat from oxidative degradation. In contrast, the superiority of ARs over even the highest concentration of sodium ascorbate (0.10%) was consistent throughout all stages of lipid oxidation, proving to successfully restrain the formation of primary oxidation products, such as hydroperoxides, and their decomposition into secondary products, such as aldehydes. The fact that the lowest tested concentration of ARs (0.01%) proved to be equal or even more effective than sodium ascorbate at 0.10% brings important implications for the food industry. Indeed, it not only enables the replacement of a synthetic antioxidant with a natural one making the label “cleaner” but it also cuts the doses normally used tenfold, with an important economic implication. However, sodium ascorbate at 0.10% promoted greater color stability, protecting the pigments responsible for the red color from degradation, while ARs proved to be ineffective at both concentrations. These results collectively suggest that ARs, at both tested concentrations, exhibit a noteworthy efficacy to delay lipid oxidation in minced-meat models, ultimately positioning them as promising natural candidates for enhancing the oxidative stability of meat and meat products. However, it will be necessary to investigate the ability of ARs to preserve the microbiological safety of the meat and protect it from spoilage together with their impact on organoleptic characteristics.

## Figures and Tables

**Figure 1 antioxidants-13-00930-f001:**
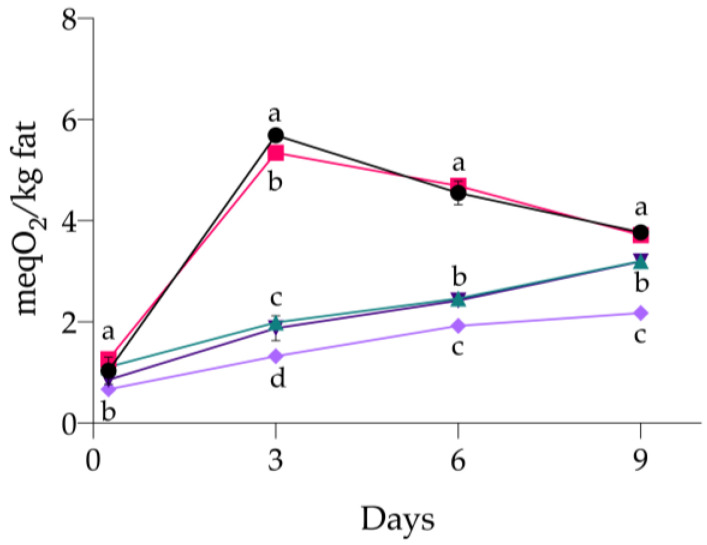
Effect of different treatments on the formation of lipid hydroperoxides (meqO_2_/kg fat) in minced-meat model systems during their storage at 4 °C for 9 days. (•) C_NEG, negative control; (∎) C_POS+, samples added with 0.01% sodium ascorbate; (▲) C_POS++, samples added with 0.10% sodium ascorbate; (▼) AR01, samples added with 0.01% ARs; (◆) AR02, samples added with 0.02% ARs. Data are presented as mean ± standard deviation (n = 3). Some error bars lie within data points. Different letters indicate means between the different treatments within the same day significantly different at *p* < 0.05.

**Figure 2 antioxidants-13-00930-f002:**
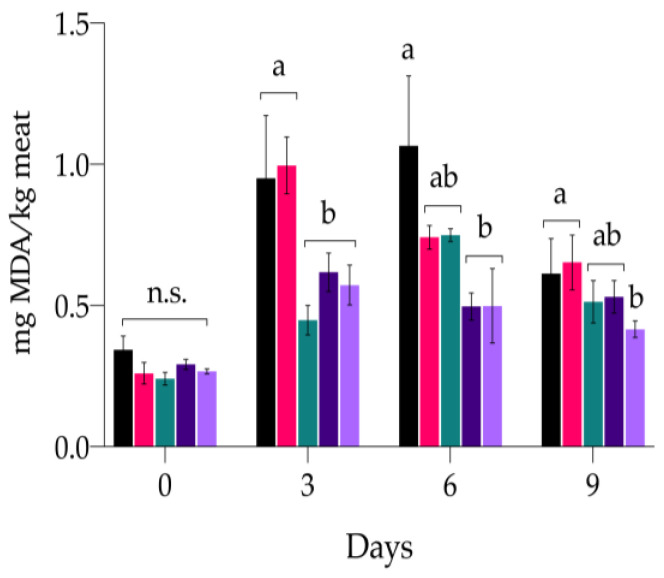
Effect of different treatments on TBARS (mg MDA/kg meat) in minced-meat model systems during their storage at 4 °C for 9 days. (∎) C_NEG, negative control; (∎) C_POS+ samples added with 0.01% sodium ascorbate; (∎) C_POS++ samples added with 0.10% sodium ascorbate; (∎) AR01 samples added with 0.01% ARs; (∎) AR02, samples added with 0.02% ARs. Data are presented as mean ± standard deviation (n = 3). Different letters indicate means between the different treatments within the same day significantly different at *p* < 0.05.

**Figure 3 antioxidants-13-00930-f003:**
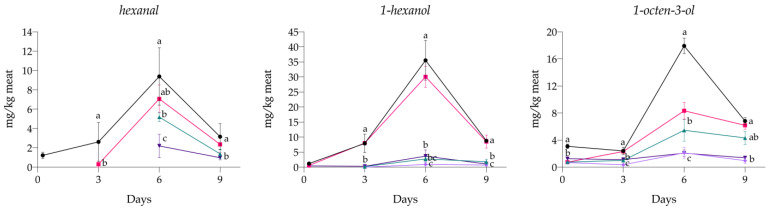
Effect of different treatments on the formation of volatile organic compounds (VOCs) from lipid oxidation in minced-meat model systems during their storage at 4 °C for 9 days. (•) C_NEG, negative control; (∎) C_POS+, samples added with 0.01% sodium ascorbate; (▲) C_POS++, samples added with 0.10% sodium ascorbate; (▼) AR01, samples added with 0.01% ARs; (◆) AR02, samples added with 0.02% ARs. Data are presented as mean ± standard deviation (n = 3). Some error bars lie within data points. Different letters indicate means within each day (comparison between different treatments) significantly different at *p* < 0.05. Where no letters are displayed, no significant differences were detected (*p* > 0.05).

**Figure 4 antioxidants-13-00930-f004:**
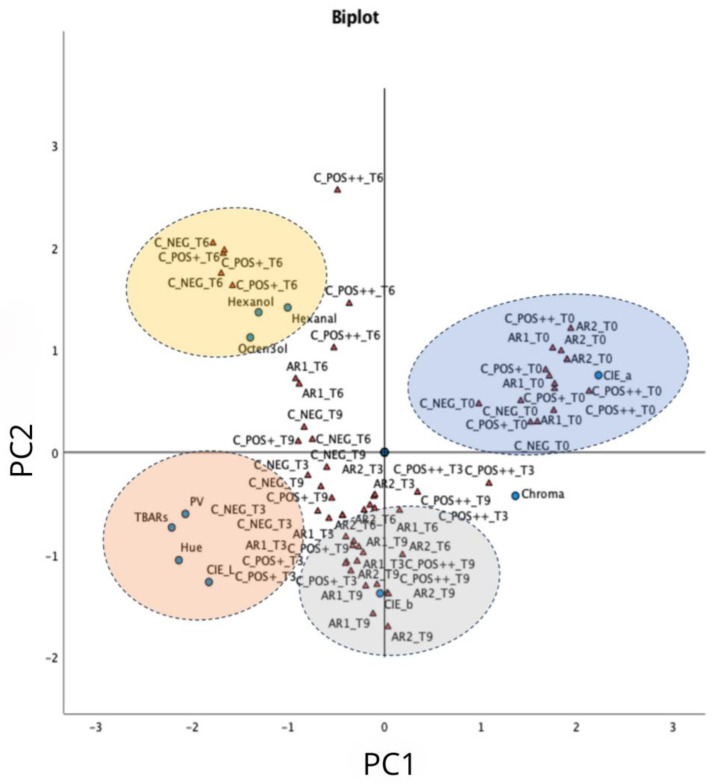
Principal component analysis (PCA) Bi-plot of hydroperoxides (PV), thiobarbituric acid reactives (TBARS), volatile organic compounds (VOCs), and CIELAB color values (*L**, *a**, *b**, *C*_ab_*, and *h_ab_*) (blue circles) and treatment time (red triangles) in minced-meat model systems (C_NEG, C_POS+, C_POS++, AR01, AR02).

**Table 1 antioxidants-13-00930-t001:** CIELAB values *L**, *a**, and *b** and calculated parameters of chroma (*C*_ab_*), and hue angle (*h_ab_*) in minced beef meat stored at 4 °C in darkness for 9 days. C_NEG, negative control; C_POS+, samples added with 0.01% sodium ascorbate; C_POS++, samples added with 0.10% sodium ascorbate; AR01, samples added with 0.01% ARs; and AR02, samples added with 0.02% ARs. Results of ANOVA and Tukey’s post hoc test are also reported.

	C_NEG	C_POS+	C_POS++	AR01	AR02	Sig.
** *L** **						
Day 0	52.05 ± 1.23 ^b^	52.48 ± 0.91 ^b^	52.44 ± 0.90 ^ab^	51.68 ± 0.97 ^c^	51.23 ± 0.91 ^b^	N.S.
3	57.25 ± 1.21 ^aA^	57.9 ± 1.63 ^aA^	53.66 ± 0.96 ^aB^	58.45 ± 0.82 ^aA^	56.89 ± 1.73 ^aA^	***
6	56.43 ± 3.11 ^aA^	56.44 ± 1.69 ^aA^	49.98 ± 2.00 ^bB^	54.53 ± 1.15 ^bA^	55.55 ± 1.19 ^aA^	***
9	57.10 ± 2.30 ^aA^	56.47 ± 1.63 ^aA^	54.07 ± 1.97 ^aB^	56.53 ± 1.68 ^aA^	56.88 ± 1.16 ^aA^	*
Sig.	***	***	**	***	***	
** *a** **						
Day 0	10.29 ± 0.61 ^a^	10.46 ± 0.48 ^a^	10.58 ± 0.58 ^a^	10.24 ± 0.48 ^a^	10.28 ± 0.47 ^a^	N.S.
3	5.45 ± 0.51 ^cC^	7.12 ± 0.95 ^bB^	10.05 ± 1.43 ^abA^	6.02 ± 0.92 ^bcBC^	5.56 ± 0.32 ^cC^	***
6	5.47 ± 0.49 ^cB^	5.63 ± 0.63 ^cB^	8.66 ± 0.69 ^bA^	5.45 ± 0.36 ^cB^	5.91 ± 0.46 ^cB^	***
9	7.06 ± 0.52 ^bB^	7.06 ± 0.41 ^bB^	8.61 ± 0.96 ^bA^	6.87 ± 0.73 ^bB^	7.46 ± 0.32 ^bB^	***
Sig.	***	***	**	***	***	
** *b** **						
Day 0	14.25 ± 0.31 ^bB^	15.11 ± 0.84 ^abB^	15.44 ± 0.79 ^A^	14.38 ± 0.82 ^bB^	14.18 ± 0.59 ^bB^	*
3	14.39 ± 0.43 ^bB^	15.52 ± 0.93 ^abA^	15.61 ± 0.55 ^A^	14.92 ± 0.61 ^bAB^	14.10 ± 0.58 ^bB^	**
6	14.80 ± 0.85 ^b^	14.29 ± 0.94 ^b^	14.64 ± 1.85	13.95 ± 0.85 ^b^	14.65 ± 0.29 ^b^	N.S.
9	16.10 ± 0.68 ^a^	16.11 ± 0.56 ^a^	15.96 ± 1.29	16.23 ± 1.01 ^a^	16.68 ± 0.36 ^a^	N.S.
Sig.	***	*	n.s.	***	***	
** *C*_ab_* **						
Day 0	17.58 ± 0.39 ^aB^	18.38 ± 0.91 ^aA^	18.72 ± 0.92 ^A^	17.67 ± 0.52 ^aB^	17.61 ± 0.43 ^aB^	*
3	15.40 ± 0.54 ^bC^	17.08 ± 1.20 ^bAB^	18.59 ± 1.24 ^A^	16.10 ± 0.87 ^bBC^	15.16 ± 0.60 ^bC^	***
6	15.78 ± 0.85 ^bAB^	15.36 ± 1.07 ^cAB^	17.03 ± 1.81 ^A^	14.98 ± 0.89 ^bB^	15.81 ± 0.38 ^bAB^	*
9	17.58 ± 0.79 ^a^	17.58 ± 0.65 ^ab^	18.14 ± 1.56	17.63 ± 1.12 ^a^	16.93 ± 1.35 ^a^	N.S.
Sig.	***	***	n.s.	***	***	
** *h_ab_* **						
Day 0	54.18 ± 1.85 ^c^	55.25 ± 1.08 ^c^	55.58 ± 1.11 ^c^	54.50 ± 2.62 ^b^	54.25 ± 2.03 ^c^	N.S.
3	69.26 ± 1.44 ^aA^	65.42 ± 1.96 ^bB^	57.36 ± 2.74 ^bcC^	68.11 ± 2.37 ^aAB^	68.47 ± 1.13 ^aAB^	***
6	69.69 ± 1.80 ^aA^	68.51 ± 1.41 ^aA^	59.21 ± 2.83 ^abB^	68.65 ± 0.99 ^aA^	68.02 ± 1.37 ^abA^	***
9	66.34 ± 1.09 ^bA^	66.31 ± 0.86 ^abA^	61.67 ± 1.33 ^aB^	67.07 ± 1.79 ^aA^	65.91 ± 0.73 ^bA^	***
Sig.	***	***	***	***	***	

Results are reported as mean ± standard deviation (n = 3). Different lowercase letters indicate means statistically different at *p* < 0.05 between different days of storage within the same treatment. Different uppercase letters indicate means statistically different at *p* < 0.05 within the same day between the different treatments. Sig. = significance; n.s./N.S. = not significant; * = *p* < 0.05; ** = *p* < 0.01; *** = *p* < 0.001.

## Data Availability

All the data supporting this research article are contained within the paper. Raw data will be made available by the authors upon reasonable request.

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
