# Peer review of "Effect of Alkylresorcinols Isolated from Wheat Bran on the Oxidative Stability of Minced-Meat Models as Related to Storage"

_antioxidants, 2024, doi:10.3390/antiox13080930_

Round 1

Reviewer 1 Report

Reviewed manuscript entitled “Effects of alkylresorcinols isolated from wheat bran on the oxidative stability of beef patties as related to storage” prepared by Carolina Cantele, Giulia Potenziani, Ambra Bonciolini, Marta Bertolino and Vladimiro Cardenia show very interesting data about antioxidants extracted from wheat bran. The Authors identified obtained compounds and used them as antioxidants in beef patties. All necessary determinations were made and the results presented and their discussion are summarized in the conclusions. Several corrections and additions will improve this manuscript.

Several corrections and additions will improve this manuscript.

 ·       It would be of great benefit to this publication to show at least sample chemical formulae of the compounds studied and to mark the fragmentation into characteristic ions.

·       What was the extraction yield of of alkylresorcinols from wheat bran?

·       In point 2.3 Characterization and Quantification of ARS the Authors described the method of GC/MS used rather for determination of the qualitative composition. But the addition of internal standard suggest readers that it was the content of alkylresorcinols analysed. What is true in this point? The results of quantitative analysis are not presented in manuscript. Why?

·       Figures 1-3: The phrase “different treatments” used in the caption is very general. The caption should contain more detailed information, without the need to look for these conditions in the methodology.

·       Figure 2: The symbols shown in the caption refer to line graphs rather than bar charts. The colors of the bars should be described in the legend of the graph.

Author Response

Response to Reviewer 1 Comments

1. Summary

Reviewed manuscript entitled “Effects of alkylresorcinols isolated from wheat bran on the oxidative stability of beef patties as related to storage” prepared by Carolina Cantele, Giulia Potenziani, Ambra Bonciolini, Marta Bertolino and Vladimiro Cardenia show very interesting data about antioxidants extracted from wheat bran. The Authors identified obtained compounds and used them as antioxidants in beef patties. All necessary determinations were made and the results presented and their discussion are summarized in the conclusions. Several corrections and additions will improve this manuscript.

We would like to thank the Reviewer for appreciating this manuscript and for taking the time to review it. Please find the detailed responses below and the corresponding revisions/corrections highlighted in red in the re-submitted files.

2. Questions for General Evaluation

Reviewer’s Evaluation

Response and Revisions

Does the title describe the article’s topic with sufficient precision?

Yes

We thank the Reviewer

Does the introduction provide a comprehensive yet concise overview about the state of knowledge in the area of research?

Yes

We thank the Reviewer

Is the research design appropriate and are the methods adequately described?

No.
One method must be corrected.

We thank the Reviewer for this comment. Please find the response in the point-by-point responses to Comments and Suggestions for Authors (comment/response 3)

Are the results presented clearly and in sufficient detail, are the conclusions supported by the results and are they put into context within the existing literature?

Yes

We thank the Reviewer

Are all of the cited references relevant to the research?

Yes

We thank the Reviewer

Does this article provide a relevant contribution to the scientific discussion of this topic?

Yes

We thank the Reviewer

Is it necessary to include study limitations in the discussion?

No.
It is not necessary to include study limitations in the Discussion.

We thank the Reviewer for this comment. However, study limitations have been included in the paper according to the other Reviewer’s suggestions.

3. Point-by-point response to Comments and Suggestions for Authors

Comment 1: It would be of great benefit to this publication to show at least sample chemical formulae of the compounds studied and to mark the fragmentation into characteristic ions.

Response 1: We thank the Reviewer for this suggestion, which surely improves our paper. We prepared a Supplementary Material including the information required. Please, find it attached with the revised paper in the MDPI procedure.

Comment 2: What was the extraction yield of alkylresorcinols from wheat bran?

Response 2: Thank you for pointing this out. The extraction yield of alkylresorcinols from wheat bran has been added in the manuscript. Please find the modified version at lines 293-295.

Comment 3: In point 2.3 Characterization and Quantification of ARS the Authors described the method of GC/MS used rather for determination of the qualitative composition. But the addition of internal standard suggest readers that it was the content of alkylresorcinols analysed. What is true in this point? The results of quantitative analysis are not presented in manuscript. Why?

Response 3: We are grateful to the Reviewer for this comment, which allowed us to notice the poor quality in the writing of the analytical method and the lack of the information on the quantification of ARs, greatly improving the manuscript. We revised Section 2.3 “Characterization and Quantification of ARs”, making the whole paragraph clearer and more understandable. The qualitative composition of ARs was carried out by GC/MS while the quantitative analysis by GC/FID by internal standard method using 5a-cholestan-3b-ol as internal standard. Please, find the modification particularly at lines 140-141 and 154-158.

Comment 4: Figures 1-3: The phrase “different treatments” used in the caption is very general. The caption should contain more detailed information, without the need to look for these conditions in the methodology.

Response 4: Thank you, we agree. The caption of Figures 1-3 have been revised accordingly. Please, find the modifications at Lines 347-353 (Figure 1), Lines 388-392 (Figure 2), and Lines 433-440 (Figure 3).

Comment 5: Figure 2: The symbols shown in the caption refer to line graphs rather than bar charts. The colors of the bars should be described in the legend of the graph.

Response 5: Thank you for pointing this out, we changed the legend accordingly (Lines 388-392).

4. Response to Comments on the Quality of English Language

Comment: English language fine. No issues detected.

Response 1: We thank the Reviewer.

Reviewer 2 Report

This is an interesting research article with adequate novelty. Some points should be addressed.

-The authors could include subheadings (Background, Methods, Results, Conclusions) in the Abstract to be more readable.

- The Introduction and the Methods section are well-written. The authors should only try to add some more information in section 2.11 concerning the statistical analysis of their results.

- The size of Figures 1 and 2 should be increased a bit.

- The resolution of Figure 3 should be improved.

- The size and the resolution of Figure 4 should be increased.

- The paragraph in lines 502-504 should be enriched by adding more data and a comparison analysis with the currently available data in the literature.

-  The paragraph in lines 505-554 is too long and it should be split into two smaller paragrphs.

- At the end of the Discussion section the authors should a new paragraph reporting the strengths and the limitations of their research study.

- the last sentence in lines 651-653 of the Conclusion section is not easily understood and it needs rephrasing.

This is an interesting research article with adequate novelty. Some points should be addressed.

-The authors could include subheadings (Background, Methods, Results, Conclusions) in the Abstract to be more readable.

- The Introduction and the Methods section are well-written. The authors should only try to add some more information in section 2.11 concerning the statistical analysis of their results.

- The size of Figures 1 and 2 should be increased a bit.

- The resolution of Figure 3 should be improved.

- The size and the resolution of Figure 4 should be increased.

- The paragraph in lines 502-504 should be enriched by adding more data and a comparison analysis with the currently available data in the literature.

-  The paragraph in lines 505-554 is too long and it should be split into two smaller paragrphs.

- At the end of the Discussion section the authors should a new paragraph reporting the strengths and the limitations of their research study.

- the last sentence in lines 651-653 of the Conclusion section is not easily understood and it needs rephrasing.

Author Response

Response to Reviewer 2 Comments

1. Summary

This is an interesting research article with adequate novelty. Some points should be addressed.

We would like to thank the Reviewer for appreciating this manuscript and for taking the time to review it. Please find the detailed responses below and the corresponding revisions/corrections highlighted/in track changes in the re-submitted files.

2. Questions for General Evaluation

Reviewer’s Evaluation

Response and Revisions

Does the title describe the article’s topic with sufficient precision?

Yes

We thank the Reviewer

Does the introduction provide a comprehensive yet concise overview about the state of knowledge in the area of research?

Yes

We thank the Reviewer

Is the research design appropriate and are the methods adequately described?

Yes

We thank the Reviewer

Are the results presented clearly and in sufficient detail, are the conclusions supported by the results and are they put into context within the existing literature?

Yes

We thank the Reviewer

Are all of the cited references relevant to the research?

Yes

We thank the Reviewer

Does this article provide a relevant contribution to the scientific discussion of this topic?

Yes

We thank the Reviewer

Is it necessary to include study limitations in the discussion?

Yes

We thank the Reviewer for this comment. Study limitations have been included in the paper. Please, find details of this in the point-by-point response to Comments and Suggestions (comment/response 8).

3. Point-by-point response to Comments and Suggestions for Authors

Comment 1: The authors could include subheadings (Background, Methods, Results, Conclusions) in the Abstract to be more readable

Response 1: We thank the Reviewer for this. Unfortunately, in agree with the journal guidelines we cannot add subtitles. Specifically, it is stated: "The abstract should be a single paragraph and should follow the style of structured abstracts, but without headings: 1) Background […]; 2) Methods […]; 3) Results […]; 4) Conclusion […] " (https://www.mdpi.com/journal/antioxidants/instructions).

Comment 2: The Introduction and the Methods section are well-written. The authors should only try to add some more information in section 2.11 concerning the statistical analysis of their results.

Response 2: We agree with the Reviewer. Additional information about the application of Shapiro-Wilk and Levene’s tests was added. Therefore, please find the modified version of Section 2.11 at Lines 270-276.

Comment 3-4-5: The size of Figures 1 and 2 should be increased a bit. The resolution of Figure 3 should be improved. The size and the resolution of Figure 4 should be increased.

Response 3: We thank the Reviewer for pointing out this, the size and resolution of all figures has been increased.

Comment 6: The paragraph in lines 502-504 should be enriched by adding more data and a comparison analysis with the currently available data in the literature.

Response 6: We think the Reviewer misinterpreted that paragraph, which was actually just a connecting sentence between the previous paragraph (in which the extract profile and its antioxidant properties are discussed and compared with data in the literature) and the next one (in which the results obtained from the beef patties are discussed). However, we agree that it may not be clear, so we have merged it with the previous paragraph and modified the sentence slightly (Lines 499-502).

Comment 7: The paragraph in lines 505-554 is too long and it should be split into two smaller paragraphs.

Response 7: The paragraph has been split in two smaller paragraphs as required (Lines 503-519 and Lines 520-552).

Comment 8: At the end of the Discussion section the authors should a new paragraph reporting the strengths and the limitations of their research study.

Response 8: We agree with the Reviewer, and we thank them for highlighting this weakness in our manuscript. We included a new paragraph accordingly. Please, find it at Lines 630-644.

Comment 9: the last sentence in lines 651-653 of the Conclusion section is not easily understood and it needs rephrasing.

Response 9: Thank you, we promptly corrected the sentence that was actually missing a piece (Line 664-666).

4. Response to Comments on the Quality of English Language

Comment: Minor editing of English language required.

Response 1: We thank the Reviewer. Language has been improved according to the Reviewer’s suggestions.

Reviewer 3 Report

The authors made an emulsion with oil in water. However, extraction of ARs involved application of various solvents such methanol.

The question is if this product is safe for human consumption. I think that it is not and this the reason that you did not proceed with the sensory analysis. In this respect, how do you plan to address the problem and make the AR product safe for human consumption. 

In this case, the title should change to a minced meat model instead of beef patties.

Please see my comments above for table axes.

Author Response

Response to Reviewer 3 Comments

1. Summary

We would like to thank the Reviewer for taking the time to review this manuscript. Please find the detailed responses below and the corresponding revisions/corrections highlighted in red in the re-submitted files.

2. Questions for General Evaluation

Reviewer’s Evaluation

Response and Revisions

Does the title describe the article’s topic with sufficient precision?

No

Please, find our detailed response in the point-by-point responses below.

Does the introduction provide a comprehensive yet concise overview about the state of knowledge in the area of research?

No.

The introduction is very interesting but the first two paragraphs on the impact of lipid oxidation in meat quality should be fairy shortened.

Thank you. The introduction has been shortened as required (Lines 30-54).

Is the research design appropriate and are the methods adequately described?

Yes

We thank the Reviewer

Are the results presented clearly and in sufficient detail, are the conclusions supported by the results and are they put into context within the existing literature?

No.

Please change day 0.25 to day 0. Day 0.25 is confusing. In the conclusions the last sentence is incomplete.

Thank you. Day 0.25 has been changed to day 0 (also in the figures and tables). Last sentence of the Conclusions Section has been revised (Line 664-666).

Are all of the cited references relevant to the research?

Yes

We thank the Reviewer

Does this article provide a relevant contribution to the scientific discussion of this topic?

Yes

We thank the Reviewer

Is it necessary to include study limitations in the discussion?

Yes

Study limitations have been now included in the paper. Please, find details of this in the point-by-point response to Comments and Suggestions (comment/response 1).

3. Point-by-point response to Comments and Suggestions for Authors

Comment 1: The authors made an emulsion with oil in water. However, extraction of ARs involved application of various solvents such methanol. The question is if this product is safe for human consumption. I think that it is not and this the reason that you did not proceed with the sensory analysis. In this respect, how do you plan to address the problem and make the AR product safe for human consumption.

Response 1: We thank the Reviewer for this comment. The alkylresorcinols were indeed extracted using non-food-grade solvents. However, the present work represents a preliminary study aimed to evaluate the ability of those natural compounds to counteract lipid oxidation in minced meat. Developing a green, food-grade extraction method without first confirming their effectiveness would have been counterproductive. Therefore, conducting a sensory test was not feasible at this stage. Nonetheless, on the basis of our obtained results the development of a safe isolation method of alkylresorcinols (as it has been done for many other natural antioxidants derived from plants, such as rosmarinic extract) is advisable. A new paragraph discussing this aspect (limitations of the study) has been added. Please, find this addition at Lines 630-644. We thank the Reviewer for this valuable comment, which has significantly improved the quality of our paper.

Comment 2: In this case, the title should change to a minced meat model instead of beef patties.

Response 2: The title has been changed accordingly. We thank the Reviewer for this suggestion, as it made the title more accurate.

4. Response to Comments on the Quality of English Language

Comment: Minor editing of English language required.

Response 1: We thank the Reviewer. Language has been improved according to the Reviewer’s suggestions.

Round 2

Reviewer 3 Report

All comments have been successfully addressed.

No comment